# The Order Matters: Sequential Fine-Tuning of LLaMA for Coherent Automated Essay Scoring

## Abstract

Automated Essay Scoring (AES) systems must judge interdependent discourse elements (e.g., lead, claim, evidence, conclusion), yet most approaches treat these in isolation, harming coherence and generalization. We investigate task-aware fine-tuning of LLaMA-3.1-8B for AES using parameter-efficient LoRA with 4-bit quantization and compare three training curricula: (i) Sequential (progressively fine-tuning on lead, then position, then claim, then evidence, then conclusion), (ii) Independent (task-specific models), and (iii) Randomized (shuffled multi-task). Experiments on the PERSUADE 2.0 corpus show that modeling task dependencies matters: Sequential fine-tuning yields the strongest overall results, including F1 scores of 65% (evidence) and 87% (conclusion) and corresponding accuracies of 63% and 85%, surpassing Independent training and outperforming a general-purpose LLaMA-70B baseline on conclusion despite its far larger capacity. Randomized training improves position scoring (57% F1) but is less consistent elsewhere. These findings indicate that (1) curriculum design aligned with discourse structure can materially improve AES, and (2) small, task-optimized models can be competitive with substantially larger Large Language Models (LLM), offering a practical path to scalable, cost-effective assessment. We release templates and implementation details to facilitate reproduction and future work on curriculum design for educational NLP.

## 1 Introduction

Automated Essay Scoring (AES) has become an increasingly important area of research in the field of artificial intelligence and education (Bai et al., 2022; Conijn et al., 2023; Mizumoto & Eguchi, 2023). With the rising demand for scalable and efficient feedback systems, AI-based AES models provide a promising solution for evaluating student essays in a consistent and timely manner (Mizumoto & Eguchi, 2023; Misgna et al., 2025; Ormerod et al., 2021). In the accurate assessment of different components of an essay, existing AES models struggle with ensuring fair, reliable, and generalizable performance across diverse writing styles and topics (Yang et al., 2024b). In traditional grading, humans evaluate essays holistically by considering the relationships between different sections. However, most AES models struggle to effectively capture these task dependencies within an essay, such as how the clarity of a student's position influences the strength of their claim or the effectiveness of their evidence (Misgna et al., 2025; Yamaura et al., 2023; Fink et al., 2024).

A key problem in AES is that an essay consists of multiple interdependent sections, such as the introduction, body, and conclusion. Scoring each section independently may lead to inconsistencies because some aspects of writing are inherently dependent on prior components. For example, a weak introduction can directly impact how well the claims in the body are structured, making it difficult for an AI system to fairly assess these components in isolation. Despite this, many existing AES models treat essay components separately, without considering how learning from one section might improve scoring accuracy for others (Tate et al., 2024). This limitation significantly reduces the effectiveness of AES models in providing meaningful feedback to students, as they fail to reflect the logical flow and coherence of an essay (Misgna et al., 2025; Singla et al., 2021).

Another challenge is the generalizability of AES models. Many fine-tuned models tend to overfit on training data and subsequently fail to maintain the same accuracy when scoring unseen essays (Yang et al., 2024a). This raises concerns about the reliability of these models in real-world applications because student essays vary in structure, content, and writing proficiency (Demszky et al., 2024). To address these issues, it is necessary to explore alternative fine-tuning approaches that improve both the generalizability and robustness of AES models (Uto & Okano, 2020; Ridley et al., 2020; Do et al., 2025; Yang et al., 2020).

To tackle these challenges, this study investigates four distinct fine-tuning strategies for AES using LLaMA-based models. The purpose is to determine which fine-tuning approach best captures the hierarchical nature of essay components and enhances scoring accuracy while maintaining model generalizability. The proposed fine-tuning approaches are the following: 1. Sequential Fine-Tuning, 2. Independent Fine-Tuning, 3. Randomized Fine-Tuning (Shuffled multi-task), and 4. Baseline Comparison (LLaMA 70B). By comparing these approaches, we aim to understand whether task dependencies play a crucial role in AES fine-tuning and whether sequential fine-tuning improves model performance compared to independent or mixed approaches.

Our approach presents several key advantages over traditional AES methods. We address critical limitations in coherence, generalization, scalability, and benchmarking. First, our method incorporates task dependency modeling and recognizes the natural relationships between different sections of an essay. Unlike previous AES models that evaluate writing components in isolation, our sequential fine-tuning strategy enhances coherence in scoring by leveraging these dependencies. Next, we aim to improve generalization by systematically comparing sequential and mixed fine-tuning strategies. This comparison allows us to identify the most effective approach to mitigate overfitting and ensures that our model maintains a strong performance on unseen essays. Additionally, our approach emphasizes scalability and efficiency by fine-tuning smaller LLaMA models on targeted writing tasks. This strategy enables us to achieve high performance while using significantly fewer computational resources, making AES systems more practical and accessible for real-world applications. Finally, we benchmark our fine-tuned models against LLaMA 70B to assess whether smaller, efficiently fine-tuned models can match or even surpass the performance of large-scale models. Our findings provide valuable insight into the feasibility of smaller models for AES given the high computational costs associated with deploying larger ones in educational settings.

This study makes several significant contributions to the field of AES and AI-assisted education. First, we conduct a comprehensive analysis of fine-tuning strategies by systematically comparing sequential, independent, and mixed fine-tuning approaches. This analysis provides valuable insights into how different training methods influence model performance on AES tasks. Additionally, by exploring task dependencies, we examine how learning various essay components in a specific order can affect overall scoring accuracy. This investigation offers a novel perspective on hierarchical learning in AES, emphasizing the importance of structured fine-tuning. Furthermore, we evaluate model generalizability by assessing whether different fine-tuning techniques impact a model's ability to perform well on new essay prompts. Addressing this key limitation in current AES research ensures that our findings contribute to the development of more robust and adaptable scoring models. Finally, we conduct a comparative study against the large-scale LLaMA 70B model to determine whether a smaller, fine-tuned model can achieve competitive or superior performance. This evaluation underscores the potential for cost-efficient AES systems that maintain high accuracy while reducing computational demands, making AI-driven essay scoring more accessible for real-world educational applications.

The effectiveness of AES depends not only on the quality of AI models but also on how they are fine-tuned to capture the complex structure of writing. In this paper, we aim to improve AES performance by exploring different fine-tuning strategies and assessing their impact on model accuracy, task dependencies, and generalizability. Our findings will provide valuable insights for building more reliable, scalable, and effective AES systems that enhance AI-assisted education. In the following sections, we review related work, analyze our dataset, and present a detailed methodology, including model design and mathematical formulations. Our evaluation examines performance through quantitative analysis, comparisons with SOTA baseline, and visualizations. Finally, we discuss key findings and future research directions to enhance scalable and reliable AES systems.

## 2 RELATED WORK

AES has been widely explored in recent research, particularly with the rise of Large Language Models (LLM) for text evaluation. Recent studies have examined various aspects of AES, including the reliability and validity of LLM-based scoring, the role of fine-tuning in improving performance, and the impact of structured prompting strategies. This section reviews key studies relevant to our research, highlighting their contributions and the gaps that our study seeks to address.

One of the most relevant studies is by Pack et al. (2024), which investigates the validity and reliability of LLMs for AES in the context of English language learner (ELL) writing. The authors evaluate multiple LLMs, including Google's PaLM 2, Anthropic's Claude 2, and OpenAI's GPT-3.5 and GPT-4, to assess their effectiveness in essay evaluation. Their findings highlight the variability in scoring reliability, with GPT-4 demonstrating the highest consistency. A key takeaway from this study is that LLMs exhibit fluctuations in scoring accuracy over time, which raises concerns about overfitting and generalizability—a central issue our research aims to address through fine-tuning strategies. Additionally, this study underscores the importance of aligning AI-generated scores with human ratings, a concept we incorporate into our evaluation by benchmarking fine-tuned LLaMA models against LLaMA 70B as a baseline. The discussion on prompt engineering further emphasizes that scoring accuracy can be influenced by how tasks are framed, aligning with our exploration of whether structured fine-tuning enhances model robustness and consistency.

Similarly, the study by Mansour et al. (2024) examines the effectiveness of LLMs for AES, evaluating ChatGPT and LLaMA models in both holistic and trait-based scoring. Their findings highlight several challenges, including prompt sensitivity, scoring inconsistency, and the performance gap between general-purpose LLMs and specialized AES models. This study is relevant to our research because we aim to determine whether our different fine-tuning strategies can mitigate such inconsistencies and improve model reliability. Mansour et al. also emphasize that LLMs struggle to differentiate between high- and low-quality essays. This reinforces the need for structured fine-tuning to enhance a model's ability to capture task dependencies and improve scoring precision. Furthermore, their comparison of LLM-based AES models with state-of-the-art (SOTA) AES models aligns with our purpose of assessing whether strategically fine-tuned smaller LLaMA models can match or surpass larger LLaMA 70B models in performance and efficiency.

Another closely related study by Stahl et al. (2024) explores the use of LLM prompting strategies for joint essay scoring and feedback generation. Their research investigates zero-shot and few-shot learning to determine how effectively LLMs can evaluate essays while providing meaningful feedback. One of their key findings is that combining AES with feedback generation enhances scoring performance, though the relationship between scoring quality and feedback effectiveness remains weak. While their focus is on optimizing LLM responses through structured prompting, our study extends this research by examining whether structured fine-tuning approaches can further enhance AES performance. Their study's emphasis on LLMs benefiting from structured guidance supports our hypothesis that fine-tuning can improve scoring consistency and generalization. Furthermore, their work highlights the trade-offs between scoring accuracy and feedback generation, which aligns with our broader goal of developing a scalable, fair, and explainable AES system.

The paper "How well can LLMs Grade Essays in Arabic?" by Ghazawi & Simpson (2025) is relevant to our study as it explores the effectiveness of state-of-the-art LLMs in AES on Arabic-language essays. The authors assess multiple LLMs, including ChatGPT, LLaMA, Aya, Jais, and ACEGPT, using zero-shot, few-shot, and fine-tuning approaches. Their findings show performance gaps between LLMs and smaller, specialized AES models in handling linguistic complexities and tokenization challenges in Arabic. The study then demonstrates how prompt engineering and instruction-following capabilities impact AES performance, showing that carefully structured prompts can enhance model accuracy. This work is highly relevant to our research as we investigate the impacts of different fine-tuning strategies on AES performance in the case of task-dependent scoring of essay components (lead, position, claim, evidence, and conclusion). While Ghazawi and Simpson examine performance of LLMs on Arabic AES, our study extends this analysis to English AES and focuses on structured fine-tuning approaches such as sequential, independent, and mixed fine-tuning. Their findings on the limitations of LLMs in automated grading reinforce our motivation to evaluate whether fine-tuning can improve scoring consistency and mitigate model instability. Furthermore, their comparison of LLMs with smaller, domain-specific models (e.g., BERT-based systems) aligns

with our goal of benchmarking fine-tuned LLaMA models against a stronger baseline (LLaMA 70B) to determine whether smaller, task-optimized models can outperform large, generic LLMs. By addressing similar challenges in different linguistic contexts, this paper provides valuable insights into the role of fine-tuning, prompt engineering, and model specialization in AES and supports our efforts to enhance the reliability and scalability of AI-powered essay grading systems.

Together, these studies provide crucial information concerning the challenges and opportunities in LLM-based AES. They highlight key concerns such as model reliability, prompt sensitivity, and the limitations of purely in-context learning approaches. Our research builds on these findings by exploring three distinct fine-tuning strategies for LLaMA-based AES models, systematically evaluating their impact on scoring accuracy, generalizability, and task dependency modeling. By bridging the gaps identified in the previous works, we aim to develop a robust and scalable AES framework that enhances AI-assisted education.

## 3 DATASET DESCRIPTION

To train and evaluate our AES models, we utilize the PERSUADE 2.0 [1] corpus dataset, a large-scale dataset designed for assessing written argumentation Crossley et al. (2024). This dataset comprises over 25,000 argumentative essays written by 6th to 12th-grade students in the United States, covering 15 different prompts across two writing tasks: independent writing and source-based writing. Each essay in the dataset is annotated with detailed discourse elements, including position, claims, evidence, counterclaims, rebuttals, and conclusions, making it highly suitable for fine-tuning AES models. The dataset includes holistic essay scores, which assess overall writing quality and effectiveness ratings for individual discourse elements. By leveraging this dataset, our study aims to develop a more context-aware AES model that accurately evaluates essays while capturing interdependencies between different components of an argument.

## 4 METHODOLOGY: LEARNING DISCOURSE-AWARE REPRESENTATIONS VIA FINE-TUNING CURRICULA

Our core objective is to investigate how different supervised fine-tuning strategies can induce representations in a LLM that are sensitive to the inherent dependencies among discourse components in argumentative essays. We frame Automated Essay Scoring (AES) not merely as a classification task, but as a problem of learning discourse-aware representations. To this end, we systematically compare three distinct training curricula for adapting a pre-trained LLM to evaluate five key essay components: lead, position, claim, evidence, and conclusion. Our experiments are designed to test the hypothesis that a curriculum mirroring the logical flow of an essay yields superior representations compared to task-agnostic or isolated training paradigms.

### 4.1 MODEL AND PARAMETER-EFFICIENT ADAPTATION

We use LLaMA-3.1-8B as our base model, which has been pre-trained on a massive corpus of text using a self-supervised objective. To adapt this model to the supervised AES task efficiently, we employ Low-Rank Adaptation (LoRA) (Hu et al., 2021). Instead of updating the full weight matrices $W_0 \in \mathbb{R}^{d \times k}$ of the transformer, LoRA injects trainable, low-rank matrices $A \in \mathbb{R}^{d \times r}$ and $B \in \mathbb{R}^{r \times k}$ into the model's self-attention layers, where the rank $r \ll \min(d, k)$. The forward pass is modified as:

$$h = W_0 x + \Delta W x = W_0 x + BAx \tag{1}$$

This approach dramatically reduces the number of trainable parameters, allowing us to learn task-specific representations without incurring the computational cost of full fine-tuning or risking catastrophic forgetting of the model's powerful pre-trained knowledge.

To make training feasible on a single A100 GPU, we further optimize the process by leveraging 4-bit quantization (specifically, NF4) via the Unsloth library. This reduces the model's memory footprint while maintaining near-original performance. Training is managed using the Hugging Face TRL `SFTTrainer`, which is designed for supervised fine-tuning of LLMs on instruction-formatted data.

---

[1] Dataset URL: https://github.com/scrosseye/persuade_corpus2.0

## 4.2 PROBLEM FORMULATION

Let the PERSUADE 2.0 dataset be a collection of tuples $(c, y, t)$, where $c$ is the text of a discourse component, $y$ is its effectiveness label (e.g., "Effective," "Adequate," "Ineffective"), and $t \in T = \{\text{Lead, Position, Claim, Evidence, Conclusion}\}$ is its component type. Our goal is to learn a mapping $f_\theta : (c, t) \rightarrow y$ parameterized by $\theta$. The parameters are initialized from the pre-trained LLaMA-3.1-8B model, $\theta_0$, and updated with LoRA adapters, $\Delta\theta$. The central question is how the training curriculum over the set of tasks $T$ influences the quality of the learned representations, as measured by downstream classification performance.

## 4.3 INVESTIGATING TRAINING CURRICULA FOR REPRESENTATION LEARNING

We explore three distinct curricula to train the LoRA adapters, each embodying a different hypothesis about how to best learn representations for interdependent tasks.

### 4.3.1 INDEPENDENT (SINGLE-TASK) FINE-TUNING

This strategy serves as a baseline to assess the value of shared representations. We train a separate set of LoRA adapters, $\Delta\theta_t$, for each discourse component type $t \in T$. Each model is trained independently from the base pre-trained weights $\theta_0$:

$$\theta_t = \theta_0 + \Delta\theta_t \quad \text{where} \quad \Delta\theta_t = \arg\min_{\Delta\theta} \mathcal{L}(f_{\theta_0+\Delta\theta}; D_t) \tag{2}$$

Here, $D_t$ is the subset of the data corresponding to component type $t$, and $\mathcal{L}$ is the cross-entropy loss. This approach produces specialized models but cannot leverage potential synergies or shared linguistic features across different discourse roles.

### 4.3.2 RANDOMIZED (MULTI-TASK) FINE-TUNING

In this approach, we learn a single, shared set of LoRA adapters, $\Delta\theta_{multi}$, by jointly training on all tasks. The training data is constructed by pooling all component datasets, $D_{multi} = \bigcup_{t \in T} D_t$, and shuffling them randomly. The model is optimized to minimize the loss over this mixed dataset:

$$\theta_{multi} = \theta_0 + \Delta\theta_{multi} \quad \text{where} \quad \Delta\theta_{multi} = \arg\min_{\Delta\theta} \mathcal{L}(f_{\theta_0+\Delta\theta}; D_{multi}) \tag{3}$$

This multi-task learning (MTL) paradigm encourages the model to find a common representational subspace that is beneficial for all component types, but it treats the tasks as independent and identically distributed, ignoring any sequential or hierarchical structure.

### 4.3.3 SEQUENTIAL (CURRICULUM) FINE-TUNING

This strategy, our primary focus, tests the hypothesis that modeling the logical dependencies of essay writing provides a powerful inductive bias. We fine-tune the model sequentially, following the natural writing order: Lead → Position → Claim → Evidence → Conclusion. The parameters learned from one task serve as the initialization for the next. Formally, starting with $\theta^{(0)} = \theta_0$, the model parameters are updated iteratively for $i = 1, \ldots, 5$:

$$\theta^{(i)} = \text{Train}(\theta^{(i-1)}, D_{t_i}) \tag{4}$$

where $(t_1, \ldots, t_5)$ is the ordered sequence of tasks and $\text{Train}(\theta, D)$ denotes fine-tuning the parameters $\theta$ on dataset $D$. This curriculum learning approach allows the model to progressively build more complex representations, leveraging the knowledge gained from foundational components (e.g., identifying a clear `Position`) to better evaluate dependent components (e.g., assessing the relevance of `Evidence`).

## 4.4 EXPERIMENTAL SETUP AND BASELINE

All models were fine-tuned using the AdamW optimizer (8-bit) with a learning rate of $2 \times 10^{-4}$, a weight decay of $0.01$, and a linear learning rate scheduler with 5 warm-up steps. We used a batch size of 2 per device and gradient accumulation over 4 steps, resulting in an effective batch size of 8. The maximum sequence length was capped at 2048 tokens.

Table 1: Performance of fine-tuning curricula across essay components. We report weighted F1-score (%) and accuracy (%). The best result for each component is highlighted in bold. The `baseline` is LLaMA-70B (zero-shot).

| | Lead | | Position | | Claim | | Evidence | | Conclusion | |
|---|---|---|---|---|---|---|---|---|---|---|
| **Method** | **F1** | **Acc** | **F1** | **Acc** | **F1** | **Acc** | **F1** | **Acc** | **F1** | **Acc** |
| LLaMA-8B (Base) | 14 | 9 | 16 | 10 | 13 | 7 | 20 | 12 | 16 | 12 |
| Baseline (70B) | **69** | **61** | **81** | **79** | **61** | 50 | 49 | 34 | 60 | 48 |
| Independent | 62 | 57 | 39 | 47 | 34 | 34 | 42 | 41 | 12 | 7 |
| Randomized | 7 | 6 | 57 | 43 | 44 | 31 | **65** | 57 | 80 | 71 |
| Sequential | 62 | 57 | 42 | 50 | 40 | **50** | **65** | **63** | **87** | **85** |

To contextualize the performance of our fine-tuned 8B models, we establish a powerful baseline using a general-purpose LLaMA-70B model in a zero-shot setting. This comparison allows us to evaluate whether a smaller, specialized model trained with a carefully designed curriculum can learn representations that are more effective for AES than those emerging from a much larger, untuned model.

# 5 EXPERIMENTS AND RESULTS

This section details the experimental setup, presents the performance of our models, and provides an analysis of how different fine-tuning curricula affect the learning of discourse-aware representations for Automated Essay Scoring (AES).

## 5.1 EXPERIMENTAL SETUP

We evaluate our models on the test split of the PERSUADE 2.0 corpus. Performance is measured using two standard classification metrics: Accuracy and Weighted F1-Score. The F1-score is particularly important as it provides a balanced measure of precision and recall, making it robust to potential class imbalances in the effectiveness labels.

We compare the following five models:

1. **LLaMA-70B (Zero-Shot):** A large-scale, general-purpose baseline to assess the zero-shot reasoning capabilities of a state-of-the-art LLM. We refer to this as the `Baseline` in our results.

2. **LLaMA-8B (Base):** The base LLaMA-3.1-8B model without any fine-tuning, used to establish the pre-trained performance floor.

3. **Independent:** Five separate LLaMA-8B models, each fine-tuned on a single discourse component.

4. **Randomized:** A single LLaMA-8B model fine-tuned on a randomly shuffled mixture of all five discourse component datasets (multi-task learning).

5. **Sequential:** Our proposed curriculum learning approach, where a single LLaMA-8B model is progressively fine-tuned on the components in a logical order (Lead → Position → Claim → Evidence → Conclusion).

## 5.2 RESULTS AND ANALYSIS

The comprehensive results for all models across the five essay components are presented in Table 1. These trends are further visualized in Figure 1, which illustrates the performance patterns for both F1-score and accuracy. Our analysis reveals several key findings regarding the efficacy of modeling task dependencies.

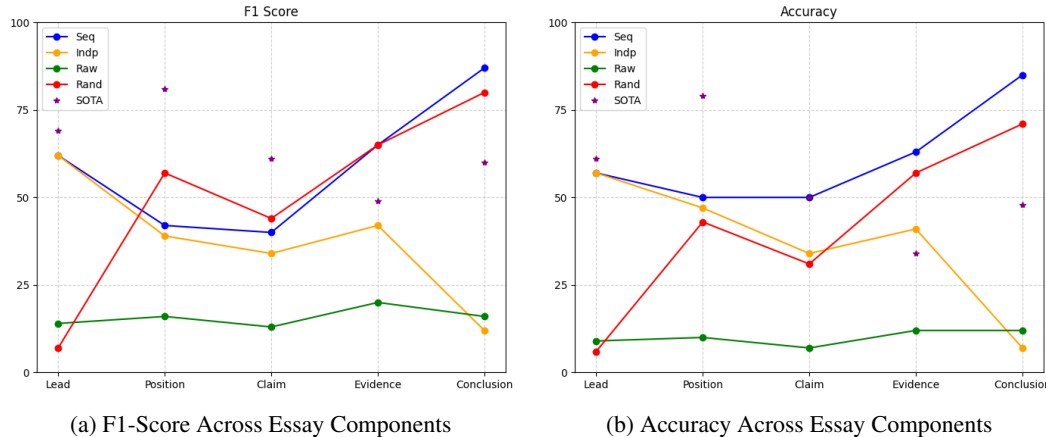

(a) F1-Score Across Essay Components   (b) Accuracy Across Essay Components

Figure 1: Comparison of F1-scores and accuracy for all fine-tuning methods across the five discourse components.

**1. Fine-Tuning is Essential for Task-Specific Adaptation.** The LLaMA-8B (Base) model performs poorly across all tasks, with F1-scores ranging from 13% to 20%. This result is expected and confirms that pre-trained models, without supervised adaptation, lack the specific representations needed for the nuanced task of AES.

**2. Sequential Curriculum Learning Yields the Strongest Overall Performance.** Our primary hypothesis is strongly supported by the results. The `Sequential` model achieves the highest or tied-for-highest F1-scores on three of the five components: `Claim` (40%), `Evidence` (65%), and `Conclusion` (87%). Its performance is particularly dominant on the most integrative components of an essay.

**3. Task Interdependence is Critical for Coherent Evaluation.** A stark contrast is visible between the `Sequential` and `Independent` models, particularly on the `Conclusion` task. While the `Sequential` model excels (87% F1), the `Independent` model catastrophically fails (12% F1). This divergence strongly implies that evaluating a conclusion effectively requires contextual representations informed by other parts of the essay.

**4. Randomized Multi-Task Learning Shows Inconsistent Benefits.** The `Randomized` (MTL) approach produces mixed results. It unexpectedly achieves the best F1-score on `Position` (57%) but performs exceptionally poorly on `Lead` (7% F1). This inconsistency suggests that while jointly learning shared features is beneficial, it is less robust than a structured curriculum.

**5. Small, Task-Aware Models Can Outperform Larger, Generalist Models.** A key finding is the competitiveness of our fine-tuned 8B models against the much larger LLaMA-70B `Baseline`. While the 70B model excels on self-contained components like `Position`, our `Sequential` 8B model significantly outperforms it on context-dependent components like `Evidence` (65% vs. 49% F1) and `Conclusion` (87% vs. 60% F1).

In summary, our results provide compelling evidence that the curriculum used for fine-tuning has a profound impact on model performance in AES, enabling smaller models to learn effective, task-specific representations that can surpass larger, general-purpose counterparts. The training loss dynamics for each fine-tuning strategy, which offer further insight into the learning process, are detailed in Appendix C.

## 6 DISCUSSION AND CONCLUSION

This study investigated the critical role of training curricula in fine-tuning LLMs for the structured task of Automated Essay Scoring (AES). Our systematic comparison of independent, multi-task,

and sequential fine-tuning strategies for LLaMA-3.1-8B revealed a clear conclusion: modeling the inherent dependencies of discourse yields substantial performance gains. The proposed sequential curriculum, which mirrors the logical flow of argumentative writing, consistently outperformed task-agnostic and isolated training paradigms, particularly on integrative components like `Evidence` and `Conclusion`. Critically, we demonstrated that a compact 8B model, when fine-tuned with a discourse-aware curriculum, can learn representations that are more effective for these complex sub-tasks than those of a much larger, general-purpose LLaMA-70B model. This finding challenges the paradigm that larger models are unilaterally better, underscoring the profound impact of task-aligned data presentation on learning efficient and specialized representations.

The results offer strong evidence that for tasks with compositional structure, the fine-tuning curriculum itself acts as a powerful inductive bias. The catastrophic failure of the independently trained model on scoring conclusions, for instance, suggests that representations for certain discourse components are deeply entangled with those that precede them. Our sequential approach provides a simple yet effective method for encouraging this knowledge transfer. The implications extend beyond AES to other structured prediction tasks in NLP, such as long-form question answering, narrative generation, and summarization, where the evaluation of one part of the text is contingent upon understanding others. Furthermore, our work provides a practical blueprint for developing smaller, cost-effective, and specialized models that are viable for real-world deployment in educational technology, offering a more scalable alternative to resource-intensive proprietary APIs.

While our findings are promising, we acknowledge certain limitations that pave the way for future work. Our analysis is situated within the context of English argumentative essays using the PER-SUADE 2.0 corpus. A crucial next step is to assess the generalizability of our curriculum-based findings to other languages, writing genres (e.g., narrative, scientific), and datasets. Future research should also explore more sophisticated training frameworks. For instance, formalizing the knowledge transfer we observed could involve continual learning approaches that explicitly mitigate catastrophic forgetting or multi-task learning schemes with structured parameter sharing, moving beyond simple sequential fine-tuning.

Perhaps the most critical future direction lies in improving model interpretability. For AES systems to transition from black-box graders to trusted pedagogical tools, they must provide transparent, actionable feedback. Integrating techniques from explainable AI (XAI), such as layer-wise relevance propagation or feature attribution methods, is essential to illuminate *why* a model assigned a particular score. Uncovering the features the model deems salient could not only build trust but also provide invaluable insights for both students and educators.

In conclusion, this work demonstrates that *how* a model is taught is as important as *what* it is taught. By aligning the fine-tuning process with the intrinsic structure of the task, we can induce more robust and efficient representations in LLMs. This curriculum-driven perspective offers a promising avenue for building more effective, interpretable, and scalable AI systems for education and beyond.

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

## A  APPENDIX

## B  PROMPT FORMATTING

To facilitate the model's understanding of argumentative essay components, we employed a standardized prompt format. This format ensures that the model receives clear, structured instructions for evaluating different sections of an essay. The example below demonstrates the template used for lead statement evaluation.

```
prompt = """
Below is an instruction for evaluating the effectiveness of a lead in an argumentative essay.
Based on the input provided, determine its effectiveness and justify your response.

### Instruction:
Evaluate the effectiveness of the lead in the given argumentative essay. It can be categorized as:
- Effective: Engaging and strongly pointing toward the position.
- Ineffective: Weak, unclear, or fails to introduce the position effectively.

---

### Input:
Essay Prompt: "{}"
Full Text Essay: "{}"
Lead: "{}"

### Response: {}
"""
```

Figure 2: Prompt Formatting Template for Lead Statement Evaluation

The structured nature of this prompt ensures that the model follows consistent input-output patterns, improving scoring accuracy and maintaining clarity across different fine-tuning strategies.

## C  TRAINING LOSS ANALYSIS

This appendix provides the training loss curves for the three fine-tuning methodologies explored in our study. These graphs offer insight into the learning dynamics of each approach and visually corroborate the performance results presented in the main paper.

### C.1  SEQUENTIAL FINE-TUNING LOSS

The training loss for the sequential fine-tuning method is shown in Figure 3. A key observation is the starting loss for each successive task. After an initial high loss on the first task (`Lead`),

the model begins each subsequent task (`Position`, `Claim`, etc.) at a significantly lower loss point. For instance, the loss at the start of the `Position` phase is much lower than the initial loss for `Lead`. This pattern provides strong evidence of positive knowledge transfer, where the representations learned from earlier discourse components serve as a highly effective initialization for later, dependent components. This efficient, curriculum-based learning directly supports the superior performance of the sequential model.

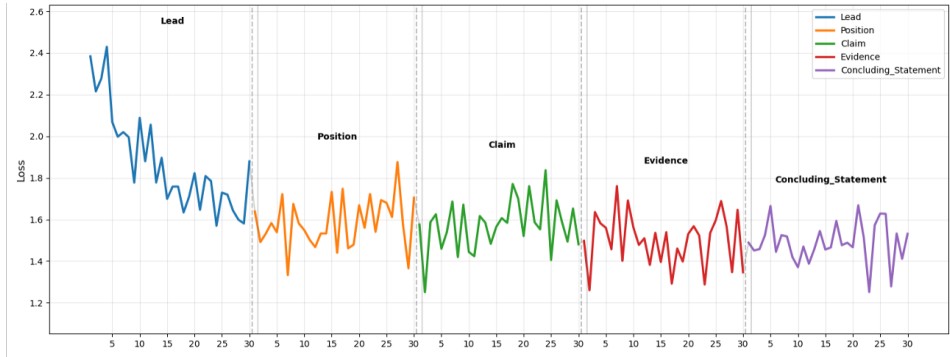

Figure 3: Training loss for the Sequential Fine-Tuning method. The model is trained progressively on each task, carrying over the learned weights. The decreasing starting loss for subsequent tasks indicates knowledge transfer.

## C.2 INDEPENDENT FINE-TUNING LOSS

Figure 4 displays the loss curves for the independent fine-tuning approach. Since each discourse component is trained using a separate model initialized from the same pre-trained LLaMA-8B checkpoint, there is no knowledge transfer between tasks. This is visually confirmed by the graph: the initial loss for each of the five tasks (`Lead`, `Position`, etc.) is consistently high (typically above 2.0). Each curve shows a standard convergence pattern, but the lack of a warm start from a related task highlights a key inefficiency of this method and helps explain its weaker performance on context-dependent components like `Conclusion`.

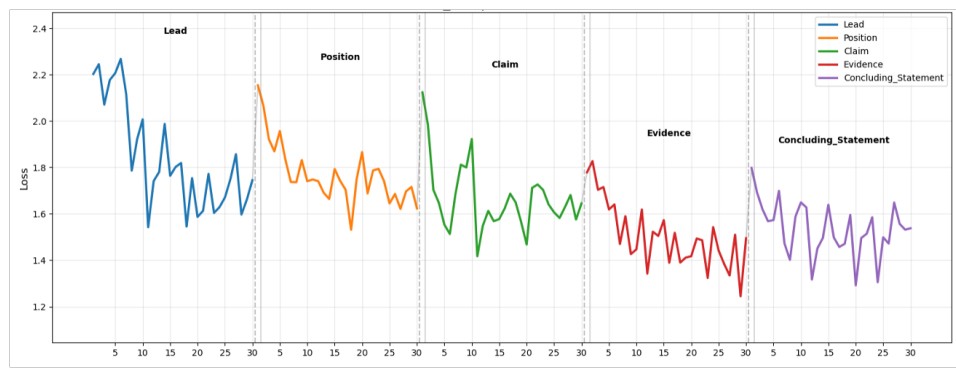

Figure 4: Training loss for the Independent Fine-Tuning method. Each colored line represents a separate model trained from scratch on a single task. Note the consistently high initial loss for each task.

## C.3 RANDOMIZED FINE-TUNING LOSS

The training dynamics for the randomized (multi-task) fine-tuning approach are presented in Figure 5. The model is trained on a shuffled mixture of all five tasks simultaneously, resulting in a single loss curve. The graph shows a rapid initial decrease in loss as the model adapts to the overall task distribution. Following this, the loss curve enters a noisy plateau, exhibiting high variance without a

smooth, monotonic decrease. This noisy behavior is characteristic of multi-task learning, where the optimization process must constantly balance competing gradients from different tasks in each batch. While the model learns a shared representation for all tasks, the lack of a structured curriculum leads to this less stable training dynamic.

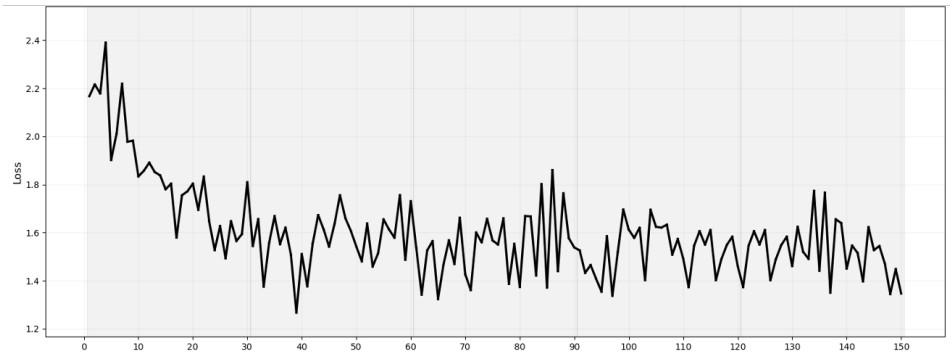

Figure 5: Training loss for the Randomized Fine-Tuning method. The single black line represents one model trained on a mixed dataset of all tasks. The high variance after initial convergence reflects the challenge of optimizing for multiple objectives simultaneously.

