# OpenReview forum: "The Order Matters: Sequential Fine-Tuning of LLaMA for Coherent Automated Essay Scoring"
_ICLR.cc/2026/Conference — ICLR 2026 Conference Withdrawn Submission_

### Official Review · Reviewer_e2G1 · 2025-10-26

**Soundness:** 3
**Presentation:** 1
**Contribution:** 2
**Rating:** 2
**Confidence:** 3

**Summary:**

The Sequential curriculum successfully captures discourse dependencies, outperforming other curricula on key components such as Evidence and Conclusion. The small fine-tuned models achieve comparable results to much larger models while greatly reducing computational cost, which enhances their suitability for educational deployment.

**Strengths:**

1.The paper highlights the importance of discourse interdependencies in essay scoring and introduces a Sequential curriculum aligned with the logical structure of essays. This approach yields strong improvements on context-dependent components such as Conclusion scoring.

2.The use of LoRA with 4-bit quantization allows efficient fine-tuning of LLaMA-8B on a single GPU, making the approach computationally practical and accessible for educational settings.

3.Comprehensive comparisons among Sequential, Independent, and Randomized curricula, together with a LLaMA-70B baseline, provide clear evidence of learning behavior and convergence patterns.

**Weaknesses:**

1.Cross-dataset and cross-genre validation are absent, as experiments rely solely on PERSUADE 2.0 and do not assess performance on narrative or multilingual data.

2.The description of LoRA hyperparameters is incomplete, omitting essential settings such as adapter rank and training seed values.

3.Model interpretability is not explored; no analysis explains why the Sequential curriculum yields superior performance, which limits trust in educational applications.

4.The paper does not test alternative curriculum sequences, preventing verification that the proposed order is truly optimal.

5.Fairness analysis is missing, with no assessment of model performance across grade levels or writing proficiencies, which may lead to bias in real-world scoring.

**Questions:**

See Weaknesses.

---

### Official Review · Reviewer_wJi4 · 2025-10-28

**Soundness:** 2
**Presentation:** 2
**Contribution:** 2
**Rating:** 2
**Confidence:** 4

**Summary:**

This paper focuses on Automated Essay Scoring (AES), addressing the limitation that most AES approaches treat interdependent discourse elements (e.g., lead, claim, evidence, conclusion) in isolation. It investigates task-aware fine-tuning of LLaMA-3.1-8B via parameter-efficient LoRA with 4-bit quantization, comparing three curricula: Sequential, Independent, and Randomized. Experiments on the PERSUADE 2.0 corpus show Sequential fine-tuning yields the strongest results, outperforming Independent training and even LLaMA-70B on conclusion. It also highlights that small, task-optimized models can compete with larger LLMs, with potential for scalable AES.

**Strengths:**

1.	The paper proposes a sequential fine-tuning strategy that aligns with the natural logical flow of argumentative essays (Lead→Position→Claim→Evidence→Conclusion) which distinguishes itself from existing "independent (single-task) fine-tuning" and "randomized (multi-task) fine-tuning" approaches.
2.	The paper accurately identifies the flaw of traditional AES models of "scoring essay components in isolation", and targets two critical industry pain points: "poor model generalizability" and "high deployment costs of large models".
3.	Through the proposed sequential fine-tuning strategy, the LLaMA-3.1-8B model outperforms the LLaMA-70B model in the zero-shot setting. This provides a new perspective for structured tasks: "task-aligned fine-tuning outperforms model scale"

**Weaknesses:**

1. The study exclusively relies on the PERSUADE 2.0 corpus and evaluates model performance only on its internal test split. No experiments are conducted on out-of-distribution datasets (e.g., ASAP). Such a single-dataset evaluation makes it difficult to distinguish between true model generalizability and potential overfitting to dataset-specific characteristics.

1. The methodological design lacks quantitative analyses to substantiate its claims. For example, the paper does not provide quantitative evidence on how different components, such as claims and evidence, influence model performance. More detailed ablation or correlation analyses are needed to demonstrate the effectiveness of the proposed approach.

2. Although the paper claims that fine-tuned smaller models are more efficient, it provides no quantitative metrics to support this statement. Key indicators such as training time, GPU memory usage, and inference speed should be reported to justify the claimed efficiency and cost-effectiveness.

3. To my knowledge, LLM-as-a-Judge frameworks have shown no significant advantage over traditional PLM-based scoring models (e.g., BERT, RoBERTa) in essay evaluation tasks. The paper mentions improved performance and efficiency but does not compare against these lighter-weight baselines. Even if direct comparison is not feasible, it would be important to include evaluations across different model sizes (e.g., LLaMA variants) or other open-source series (e.g., Qwen, DeepSeek) to verify the robustness of the proposed method.

**Questions:**

1. The paper uses the PERSUADE 2.0 dataset with over 25K essays, but it does not specify the distribution of essay quality levels (e.g., high, medium, low). Could such imbalance bias the model toward certain quality categories, and have the authors considered testing on datasets with different distributions?
2. LLaMA-70B is used only in a zero-shot setting, while LLaMA-3.1-8B is fine-tuned. If the same fine-tuning strategy were applied to LLaMA-70B, might it outperform the smaller model? Does this comparison risk underestimating the large model’s capability?
3. Sequential fine-tuning achieves a striking F1 of 87% on “Conclusion,” but the dependency among components is not analyzed. Could the authors visualize or ablate components (e.g., removing “Evidence”) to confirm how “Conclusion” depends on other parts?
4. The study reports F1 and accuracy, but metrics like score consistency and correlation with human ratings are also key in AES. Why doesn't it choose more suitable metrics like QWK, Kappa, Pearson, or Spearman scores?

---

### Official Review · Reviewer_yvUu · 2025-10-31

**Soundness:** 3
**Presentation:** 3
**Contribution:** 2
**Rating:** 4
**Confidence:** 4

**Summary:**

To enhance the consistency and generalizability of Automated Essay Scoring (AES), this study explores multiple fine-tuning strategies. Based on the LLaMA-3.1-8B model and incorporating LoRA and 4-bit quantization techniques, compare three fine-tuning approaches：Sequential, Independent, and Randomized (multi-task), on the PERSUADE 2.0 dataset. On one hand, the study confirms the crucial role of task dependencies in AES fine-tuning. On the other hand, it reveals that a curriculum design aligned with discourse structure can significantly improve AES performance: Sequential fine-tunning, which follows the logical order of essay components (Lead → Position → Claim →Evidence → Conclusion)—achieves the best results in both F1 score and accuracy, even outperforming the much larger LLaMA-70B model on context-dependent scoring tasks.

**Strengths:**

A substantive assessment of the strengths of the paper, touching on each of the following dimensions: originality, quality, clarity, and significance. We encourage reviewers to be broad in their definitions of originality and significance. For example, originality may arise from a new definition or problem formulation, creative combinations of existing ideas, application to a new domain, or removing limitations from prior results

- Integrating curriculum learning with AES,the paper propose a sequential fine-tuning strategy,the experimental results demonstrate the effectiveness of this strategy.

- The 8B model surpasses the 70B model through sequential strategy, highlighting a significant efficiency advantage, offering a practical path for cost-effective and scalable AES deployment.

**Weaknesses:**

A substantive assessment of the weaknesses of the paper. Focus on constructive and actionable insights on how the work could improve towards its stated goals. Be specific, avoid generic remarks. For example, if you believe the contribution lacks novelty, provide references and an explanation as evidence; if you believe experiments are insufficient, explain why and exactly what is missing, etc.

- Insufficient ablation experiments. The central claim regarding the critical importance of the specific order is not fully substantiated. To strengthen this argument, ablation studies should be conducted comparing the proposed sequence against plausible alternatives—such as reverse order or random permutations—to isolate the effect of order from the general benefits of curriculum learning.

- Limited scope and practical impact. (a) Generalizability: It is limited to English argumentative essays from a single corpus, leaving its efficacy for other genres or languages unexplored. (b) Practicality: The study stops at component-level classification without demonstrating how to synthesize these scores into a final, holistic essay grade, which is crucial for real-world application.

- Unexplained performance inconsistency. The results show that Sequential fine-tuning underperforms compared to the Randomized method on certain dimensions. This counter-intuitive finding warrants a deeper analysis to explain why and to clarify the boundaries of the proposed method's effectiveness.

- Lacks sufficient interpretability. The analysis of the training loss dynamics does not sufficiently establish a causal link to the efficacy of the proposed method and requires a more thorough explanation. The paper contains formatting errors, including the incorrect use of quotation marks (e.g., using closing quotes in place of opening quotes) and inconsistencies in citation style.

**Questions:**

- To strengthen the core claim, please compare your specific sequence against alternative orders (e.g., reverse). This is needed to distinguish the benefit of a discourse-aware curriculum from general curriculum learning effects.

- Why does the Sequential method perform worse than the Randomized method on certain dimensions? What information might this reveal about task dependencies?

- Whether the essay final score can be integrated as an indicator to compare the impact of several fine-tuning methods on scoring results?

- The current reliance on the PERSUADE 2.0 dataset constrains the evaluation of cross-domain adaptability.Is sequential fine-tuning universally effective across diverse essay structures?

- In order to establish a more competitive and rigorous benchmark, the inclusion of well-established proprietary models (such as GPT-4) as additional baselines is necessary.

---

### Official Review · Reviewer_UDVE · 2025-11-03

**Soundness:** 2
**Presentation:** 3
**Contribution:** 2
**Rating:** 2
**Confidence:** 3

**Summary:**

This paper presents a curriculum-based fine-tuning approach to classify the effectiveness of discourse elements in an essay.

**Strengths:**

* Assessing the effectiveness of discourse elements present in an essay is a relevant problem that can help provide useful feedback to students.
* Investigates a curriculum-based fine-tuning approach to classifying discourse elements.
* Presents a parameter-efficient approach with LoRA and quantization.
* The methodology is clearly presented and well written.

**Weaknesses:**

* Validity of assumptions: The persuade dataset has 7 elements. Why is the assumption made to exclude counterclaims and rebuttals from this task? Further, are these elements sequential in the dataset? Real-world student essay components could be related in a non-linear way, as well as with repeating claim-evidence cycles. For ex: position, claim, evidence, claim, evidence, counterclaim, evidence, new position, etc
* Motivation behind task formulation: The Persuade dataset and problem defined in the Kaggle challenge provide the full text of the entire essay, as would also be available in real-world scoring. Models, therefore, could grade essay components in context with the full text. The 1st place solution of this Kaggle competition to classify the effectiveness of discourse elements in an essay (by Team Hydrogen here: https://www.kaggle.com/competitions/feedback-prize-effectiveness/writeups/team-hydrogen-team-hydrogen-1st-place-solution) uses the full text of the essay as input with token separators for different essay components. Therefore, line 050, “AES models treat essay components separately,” is not valid.
* Improvement in AES performance: While classifying the effectiveness of discourse elements in an essay is useful, the title of the paper is focused on AES. Persuade provides an overall essay score. How well do the presented models perform on AES? Does curriculum-based tuning on discourse elements improve overall AES?
* Compare against existing SOTA methods: This problem was included in a Kaggle challenge (https://www.kaggle.com/competitions/feedback-prize-effectiveness/overview). The authors should compare with best best-performing models from the challenge.

**Questions:**

* Why is finetuning of llama-8B on conclusion worse than using the vanilla pre-trained model?
* What is the variance in results?
* The Persuade 2.0 corpus has a scoring rubric. Did the authors try including the rubric?
* The authors show training loss curves in the appendix. Can the corresponding validation loss be included and analysed for overfitting, etc?

---

### Note · Authors · 2025-11-21

I have read and agree with the venue's withdrawal policy on behalf of myself and my co-authors.